# Plant species diversity assessment and monitoring in catchment areas of River Chenab, Punjab, Pakistan

**Muhammad Azhar Ali**[1], **Muhammad Sajjad Iqbal**[1]*, **Khawaja Shafique Ahmad**[2], **Muhammad Akbar**[1], **Ansar Mehmood**[2], **Syed Atiq Hussain**[1], **Noshia Arshad**[1], **Saba Munir**[1], **Hajra Masood**[1], **Tahira Ahmad**[1], **Ghulam Muhiyuddin Kaloi**[3], **Muhammad Islam**[4]

1 Biodiversity Informatics, Genomics and Post Harvest Biology Laboratory, Department of Botany, University of Gujrat, Gujrat, Pakistan, 2 Department of Botany, University of Poonch Rawalakot, Azad Jammu & Kashmir, Pakistan, 3 National Sugarcane and Tropical Horticulture Research Institute, Thatta, Pakistan, 4 Department of Genetic Engineering and Biotechnology, Hazara University, Mansehra, Pakistan

* drsajjad.iqbal@uog.edu.pk

## Abstract

### Background

Biodiversity data is crucial for sustainable development and making decisions regarding natural resources and its conservation. The study goal was to use quantitative ecological approaches to determine the species richness and diversity of wild flora and the ultimate impact of environmental factors on vegetation dynamics.

### Methods

Quadrats having sizes of 1×1 for herbs, 5×5 for shrubs, and 10×10 m² for trees were used. Various phytosociological characteristics were investigated in association with a wide variety of environmental variables. Soil analysis based on texture, moisture, pH, electrical conductivity (EC), organic matter (OM), available potassium (K), and phosphorus (P) were examined. The existing state of vegetation along the River Chenab was assessed using SWOT analysis and a future conservation strategy was devised.

### Results

One hundred twenty different plant speies were divided into 51 families including 92 dicots, 17 monocots, 6 pteridophytes and 1 bryophyte species. Herbs accounted for 89 followed by shrubs (16 species) and trees (15 species). Correlation analysis revealed a highly positive correlation between relative density and relative frequency (0.956**). Shannon and Simpson's diversity indices elaborated that site 3 and 7 with clay loamy soil had non-significant alpha diversity and varies from site to site. Diversity analysis showed that site 10 was most diverse (22.25) in terms of species richness. The principal coordinate analysis expressed that different environmental variables including OM, soil pH, P, K, and EC affect vegetation significantly, therefore, loamy soil showed presence and dispersal of more vegetation as

**Data Availability Statement:** All relevant data are within the manuscript and its Supporting Information files.

**Funding:** The author(s) received no specific funding for this work.

**Competing interests:** The authors have declared that no competing interests exist.

compared to loam, sandy and sandy loam soils. Further, 170 ppm of available potassium had significant affect on plant diversity and distribution.

## Conclusion

Asteraceae family was found dominant as dicot while poaceae among monocot. *Adhatoda vasica* was one of the unique species and found in Head Maralla site. For evenness, site 3 had maximum value 0.971. Most of the soil represented loamy soil texture where site 2 and 4 possess high soil moisture content. SWOT analysis revealed strengths as people preferred plants for medicine, food and economic purposes. In weakness, agricultural practices, soil erosion and flooding affected the vegetation. In opportunities, Forest and Irrigation Departments were planting plants for the restoration of ecosystem. Threats include anthropogenic activities overgrazing, urbanization and road infrastructure at Head Maralla, habitat fragmentation at Head Khanki, and extensive fish farming at Head Qadirabad. Future conservation efforts should be concentrated on SWOT analysis outcome in terms of stopping illegal consumption of natural resources, restoration of plant biodiversity through reforestation, designating protected areas and multiplying rare species locally.

## Introduction

The plant diversity is one of the most important component of terrestrial ecosystems, which plays a critical role in maintaining an area's stability [1]. A diversified flora also helps in slopes stabilization, soil improvement, the buffering of weather extremes and the provision of habitat for wildlife [2]. It becomes a major topic of concern in recent decades that how to conserve an environment on sustainable basis. Moreover, it is the legitimate source of fundamental needs including food, medicine, and other products, as well as a growing public awareness of its importance in ensuring human well-being [3, 4]. In case of species diversity, the best indication is species richness which is highlighted by various environmental factors such as temperature, annual rainfall and soil composition [5]. Resultantly, it affects composition of plant communities and geographical distributionin an environment [6, 7].

Variation in plant species and community structure along with the altitude and latitude is a well-established phenomenon which affects climate [8]. It induces apparent threats and even permanent loss triggered through various elements including climate change, strong demographical growth and various anthropological activities such as urbanization, industrialization, overgrazing and deforestation [9]. Deforestation along obscured vegetation caused ecological disruption, change in soil structure due to physical weathering, and rock denudation, ultimately become reason in floods and disrupted the structure of entire natural flora [10]. This threat is particularly predominant in places where significant population and density exists, consequently it derives an increase in resource demand, and eventually appears as exploitation of available natural resources *viz.*, overgrazing, deforestation and overexploitation [11, 12].

Luckily, over the last two decades, several efforts have been started to introduce regional and global plans to prioritize emerging trends and attract investment in biological conservation [13–15]. For managing an ecosystem various tools and techniques are required to take strategic planning. SWOT (strengths, weaknesses, opportunities, and threats) analysis is an approach which can be use efficiently in the management of ecological components. It is one of the quantitative applications to evaluate the selected sites and to devise some strategic

planning [16, 17]. It encompasses comprehensively both internal and external factors of the positive and negative impact [18]. SWOT analysis can also use to quantifying the existing as well as future opportunities and threats for conservation through appropriate management strategies [19].

Subsequently, to attain the effective conservation of biodiversity, it requires the biogeographic as well as macroecological analysis of the factors and processes that determine the geographic distribution of the species [19–21]. The present studies were carried out to find phytosociological analysis and effect of environmental variables on vegetation on the riverine area of river Chenab distributing into different sites. These include parameters such as identification of species, abundance, richness and status of vegetation [22]. Further, PCoA was conducted which is a type of direct gradient analysis helps in indicating the difference among vegetation patterns under the effect of different environmental variables [23]. Findings express not only species distribution, anthropogenic effects but also suggest possible conservation measurements to ensure future conservation for sustainable ecosystem.

## Material and methods

### Ethics statement

The study was approved by the Institutional Ethics Committee through No. UOG/ASRB/Botany/02/7881. Verbal informed consent was obtained from each informant before conducting the interview process.

### Study area

River Chenab is one of the largest rivers of the Indus basin which is originated from Himachal Pradesh, India and runs down 960 km in Pakistan. There are three Head works including Head Maralla comprises of 2816 ha wetland area, Head Khanki 1200 ha while Head Qadirabad spreads over 1620 ha. It is the second largest river of Pakistan and has a large wetland area in the catchments. River Chenab enters Pakistan near the line of control by two tributaries namely Munawar Tavi and Jammu Tawi. On both sides of the river these two tributaries have an area of about 2,800 km$^2$. River Chenab enters Pakistan in the upstream of Maralla rim station with 74˚29' E and 32˚ 40'N and transverse through east to west and travel the distance of 598 km and confluence at Mithan Kot in the Indus delta. In Jammu and Kashmir region, it transverses in the mountainous slope region and flows in the plain region of Punjab after entering Pakistan. The interfluve between the Ravi and Chenab is known as "Rechna Doab" and the interfluve between Chenab and Jehlum is "Chej doab". The Chenab River is confluence with the river Jhelum just upstream of Trimmu barrage followed by river Ravi adjoining the area of Ahmadpur Siyal. The streams of three rivers combine and finally meet with remaining two rivers of Punjab such as Sutluj and Beas to form Punjnad which is also known as combine streams of five rivers that are finally confluence into the Indus River [24]. Fig 1 showed the location of the study area.

**Selection of sampling sites.** The vegetation of river Chenab was studied under bioclimatic conditions to follow changes in species composition of plants by change in soil and climatic conditions. The sampling sites including were natural and artificial lands. While catchment areas included riverine areas, canal sides and river banks which were highly disturbed by the river water. The field work was carried out during summer 2019 using phytosociological techniques. Quantitative attributes such as frequency, density and abundance of each vascular plant species were measured by using the quadrat method along the altitudinal transect. The study area of about 85 km was divided into 12 sampling sites having four quadrats. Hence, a total of 48 quadrats were used for the quantitative analysis of vegetation (Fig 2).

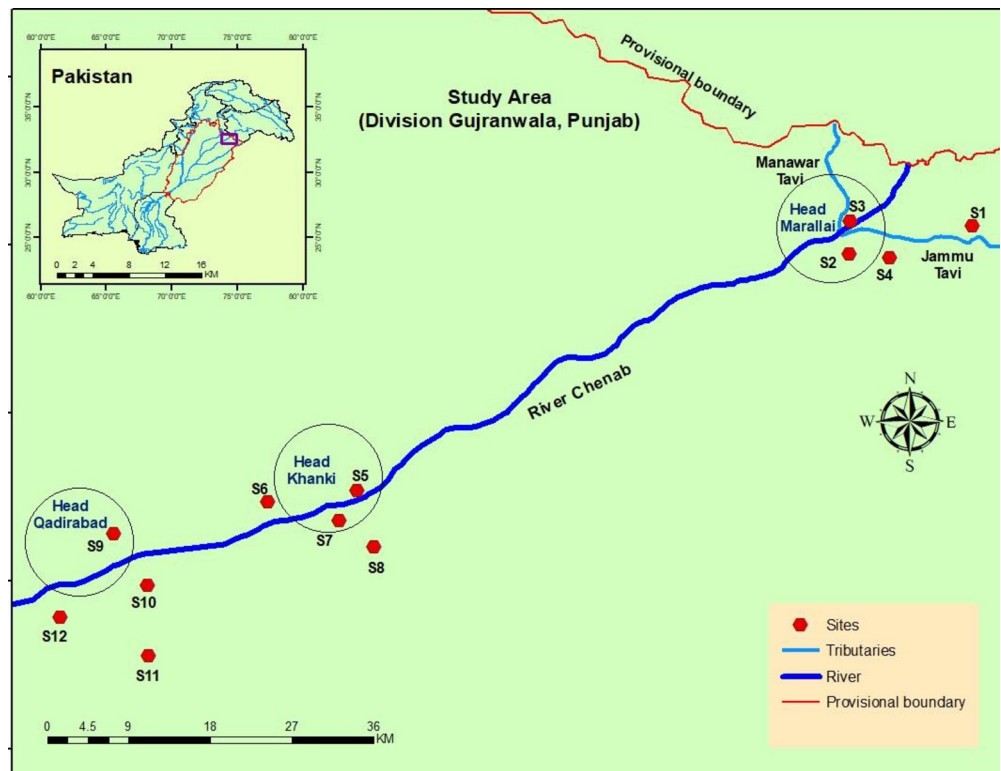

**Fig 1. Location of study area Head Maralla, Sialkot to Head Qadirabad, Mandibahudin at River Chenab, Punjab, Pakistan.**

Site 1: Jammu Tavi, Bajwat area; Jammu Tavi, Bajwat area with the coordinates of 32° 40′ 41″ N, 74°29′ 30″ E is located on the upstream side of Head Maralla. This site is near to the working boundary of India and Pakistan (Jammu and Kashmir) where it enters Pakistan.

Site 2: Head Maralla main bridge site; Head Maralla main bridge site, opposite to the old canals with the coordinates of 32° 41′ 24″N, and 74°28′46″ E. It is the main regulator reservoir opposite the old canal sides.

Site 3: Manawar Tavi, Thokar No. 4; this point is junction of three rivers Manawar Tavi, River Chenab and Jammu Tavi with the coordinates of N 32°40′12″, E 74°28′23″, located on the north upstream side of Head Maralla. Manawar Tavi originates from Azad Kashmir and enters Pakistan at the Chamb Jorian border.

Site 4: Canal colony side; Canal colony side is at west side of Head Maralla reservoir with the coordinates of N 32°30′44″, E 74°27′50″ and is at downstream.

Site 5: Bella forest, wetland area upstream of Head Khanki; it has coordinate of N 32°25′07″, E 78° 58′02″. Bella forest area located on the north side of river Chenab.

Site 6: Head Khanki main Bridge west side has coordinates of N 32°24′49″, E73° 57′ 43″. It is located on the right bank of the river side.

Site 7: Khanki village; Khanki village downstream main bridge is the water regulator reservoir with the coordinates of N 32°23′50″, E 73° 57′ 48″. Head Khanki is located on river Chenab from the 16 km downstream of Alexander Bridge and North West side of Wazirabad city.

Site 8: Lower Chenab Canal, Khanki; Lower Chenab Canal, Khanki with the coordinates of N 32°23′ 40°, E 73°58′13″. Lower Chenab Canal downstream Head Khanki.

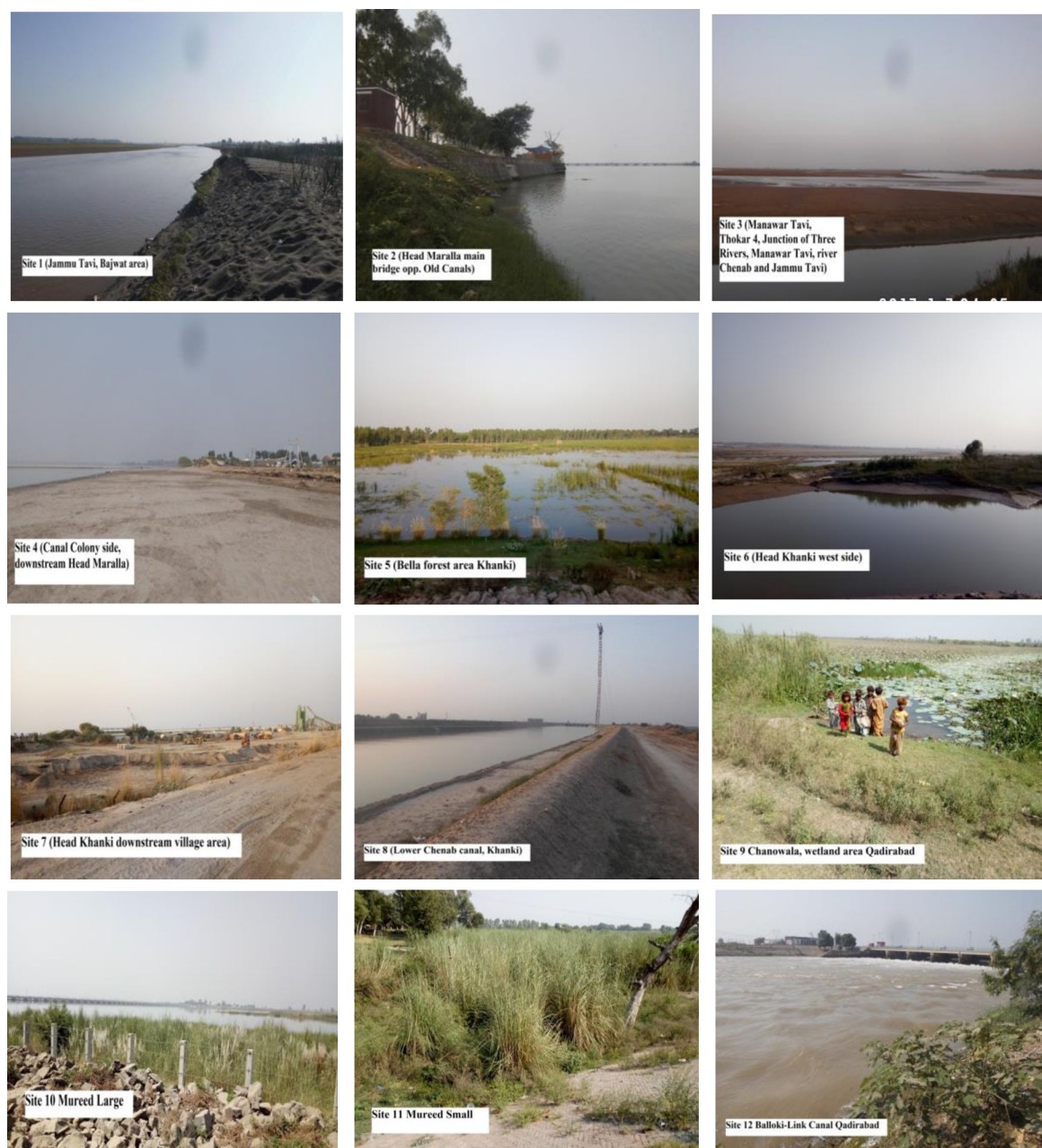

**Fig 2. Pictorial representation of selected sites around River Chenab, Punjab, Pakistan.**

Site 9: Chanowala, wetland area Qadirabad; Chanowala, wetland area Qadirabad with the coordinates of N 32°19′06″, E 73°41′13″ Chanowala side is located on the upstream side of Head Qadirabad.

Site 10: Mureed large; Mureed large with the coordinates of N 32°19′06″, E 73°41′46″ Mureed large site located on the south side of the Head Qadirabad, river Chenab.

Site 11: Mureed small; Mureed small with the coordinates of N 32°18′41″, E 73°41′59″ Mureed small site is located on the east side of Qadirabad Balloki Link canal side.

Site 12: Balloki-Link Canal Qadirabad; Balloki-Link Canal Qadirabad with the coordinates of N 37˚17′32″, E 73˚ 41′25″. Kotli downstream of Head Qadirabad is located on the west side of Qadirabad Balloki Link Canal.

**Data collection.**   Random way was used to sample the area with quadrats of different sizes for herb, shrub and trees. The use of quadrat method along the transects are the best method to assess vegetation analysis in the different landscapes [25–27]. The following steps are sequentially met during vegetation sampling. The whole study area was divided into 12 sites for quantitative sampling as shown in Fig 2. Easily accessible, mature and disturbed vegetation was sampled through quadrats at each site. Simple rope, rod and measuring inch tape were used to make quadrats. The typical quadrats size used in this study were 1×1m for herbs, 5×5 m for shrubs and 10 ×10 m for trees. At each site three quadrats on four different points were used to collect floristic data by sampling the vegetation in three layers i.e., herbs, shrubs and trees. In the first layer, the entire herbaceous flora were < 1 m in height; in the shrub layer, all of the woody flora with 1 and 5 m height; similarly, in the tree layer, woody plants with height of greater than 5 m. Presence of all of the plant species were recorded. For herbs and shrubs the value of cover/abundance was visually estimated in the quadrats following the regulated Braun-Blanquet scale modified by Daubinmire [28, 29]. Phytosociological parameters studied at different sites *viz*., cover, composition, density, frequency, relative cover, relative frequency and relative density of the species were calculated using following formulas;

$$\text{Cover} = \frac{\text{Total intercept length of a species}}{\text{Total transact length}} \text{ x } 100$$

$$\text{Composition} = \frac{\text{Number of individuals of a species}}{\text{Total number of individuals of all species}} \text{ x } 100$$

$$\text{Frequency} = \frac{\text{Number of quadrates in which a species occurred}}{\text{Total number of quadrates sampled}} \text{ x } 100$$

$$\text{Density} = \frac{\text{Number of individuals of a species in all quadrats}}{\text{Total area sampled}} \text{ x } 100$$

$$\text{Relative cover (RC)} = \frac{\text{Total intercept length of a species}}{\text{Total intercept length of all species}} \text{ x } 100$$

$$\text{Relative frequency (RF)} = \frac{\text{Frequency of a species}}{\text{Total frequency of all species}} \text{ x } 100$$

$$\text{Relative density (RD)} = \frac{\text{Total individuals of a species}}{\text{Total number of plants of all species}} \text{ x } 100$$

Herbarium specimens of all the plant species enlisted were collected from all sites and identified through the Flora of Pakistan [30]. Plant specimens were dried, processed and mounted on the standard herbarium sheets. Herbarium specimens were then deposited to the herbarium, Department of Botany, University of Gujrat, Gujrat, Pakistan and assigned Voucher specimen numbers.

**Soil analysis.**   Soil samples were collected with the help of barrel type hand auger from each site in the depth of 0–15 cm [31]. Soil samples were initially air dried and filled in polythene bags. Samples were labeled properly to represent their site of collection with date of

collection. For analyzing physical characteristics (soil texture) and chemical characteristics (saturation, pH, EC, organic matter, available Phosphorus and available Potassium), samples were transported to the laboratory [31].

## SWOT analysis

SWOT analysis (strengths, weaknesses, opportunities and threats) were determined to synthesize groups environmental descriptors [32]. The analysis was carried out on the specific site to classify the indicator and identify the potential factors that affect the suitability of the study area (Table 5). For example, strengths indicate the area, size and presence of vegetation. A weakness indicates native species, natural disasters like flood and competition among species. Threats indicate anthropogenic factors and to some extent natural and unexpected disasters while opportunities indicate the involvement of NGOs, other departments or the people working for conservation and preservation. Some indicators (exotic and local species) can be divided to specific site and area. This technique is more instinctive given the opportunities are deliberated relative to the other descriptors.

SWOT values exhibit particular number within a specific site which varies from 100 to 1000 in terms of relative importance of weight ranges. Where 100 indicates low value whereas 1000 as an indication of the higher value of significance for particular factors in the study area. In quantifying the area quality, the importance coefficient (IC) was calculated from all indicator which reflect the overall relevancy (Table 5). To determine the individual ICs, ICi = Wi/ Wi-n; where Wi is the individual indicator weight while Wi-n is the sum of all respective indicator weight and ICi n = 1 [32]. The differences in the number, importance of individual descriptors, the subtotal descriptor weights and ICs were also recorded (Table 5).

## Statistical analysis

Minitab software v. 18 was used to find the correlation between the phytosociological parameters of 120 plant species. Further, diversity indices such as Shannon-Weiner, Simpson, evenness and richness were calculated through R packages (v.3.6.1). The data matrices were log transformed using *log10* function from the zbase package before further analysis. Shannon and Simpson diversity indices were calculated by using the function *diversity* from *vegan* package (v.2.5–6) [33]. Diversity indices were further compared with Wilcox test using *ggpubr* (https:// cran.r-project.org/web/packages/ggpubr/index.html) package to confirm the significant differences and effect of soil on diversity indices. Similarly, to analyze the plant diversity with respect to different soil types (clay, loam, sandy and sandy loam) the permutational multivariate analysis of variance (PERMANOVA) was performed based on Bray-Curtis dissimilarity matrices using *vegdist* and *adonis* function from *vegan* package. Model was selected based on P and F value for further analysis. Bray-Curtis distance matrices were calculated using transformed community data and pair-wise dissimilarities between different soil types, along with environmental data, were visualized with principal coordinate analysis (PCoA) plots, using ggplot2 package [34].

## Results

### Floristic diversity

Different types of plant species were found in the study area due to variation in environmental factors. Collection represented 120 plants with 51 families belonging to 105 genera, in which dicot families were 40 and dominant with 85 genera and 97 plant species. In monocots, three families with 13 genera and 16 plant species were documented. Likewise, few species of

**Table 1. Summary of floristic diversity of River Chenab (Head Maralla to Head Qadirabad).**

| Plant Groups | Families | Genera | Species |
|---|---|---|---|
| Dicots | 40 | 85 | 96 |
| Monocots | 4 | 13 | 17 |
| Pteridophytes | 6 | 6 | 6 |
| Bryophytes | 1 | 1 | 1 |
| Total | 51 | 105 | 120 |

pteridophytes and bryophytes were also reported from the area (Table 1). Further, angiospermic group was categorized into two groups including dicots 96 species and monocots as 17 species, with prominent families poaceae and cyperaceae (S1 Table). As far as plant habit was concerned, 89 herbs, 16 shrubs and 15 tree species were found in the study area (Fig 3).

Several quantitative parameters were studied to describe the phytosociological analysis including composition, cover, density, frequency, relative cover, relative density and relative frequency. Phytosociological analyses of the twelve sites were carried out as presented following; site 1 Jammu Tavi, Bajwat, the soil of this area was loamy soil where anthropogenic factors viz., agricultural practices and grazing pressure were the main influencer on this site (Table 2). With the help of 4 quadrats 27 plant species were recorded, out of which 12 species were perennial and the remaining 15 were annual. The dominant species of this site were herbs; *Marsilea quadrifolia*, *Azolla pinnata*, *Osmunda regalis*, *Cynodon dactylon* and *Ranunculus muricatus*. Shrubs including *Calotropis procera* and *Ipomoea carnea* while in trees *Acacia nilotica* and *Broussonetia papyrifera* were documented. Phytosociological analysis of site 4 Canal Colony side; this site bearing loamy soil which was mostly used as a picnic place, the parking area and the tourist place, therefore, human intervention was very high. Eighteen plant species were recorded from this site including 7 species of perennials as well as 11 species of annuals.

Phytosociological analysis of site 5, Bella Forest area, Head Khanki exhibit loamy soil and mostly the wetland area. This site was observed with high rate of grazing by livestock and expansion of urbanization. Thirty two species were recorded including 10 species of perennials and 22 species of annuals. Dominant species of this site were herbs (*Leucas aspera*, *Digera arvensis*, *Pontederia crassipes* and *Solanum nigrum*), followed by shrubs (*Ipomoea carnea* and

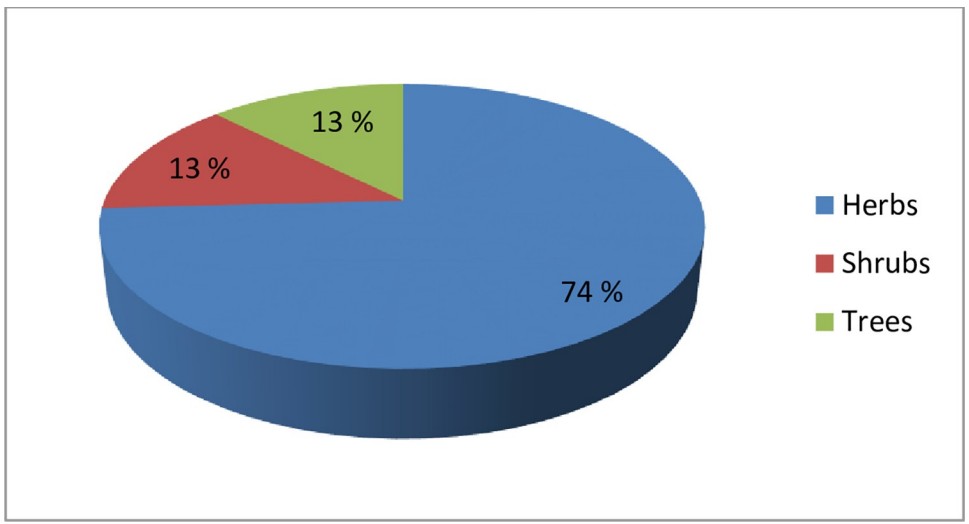

**Fig 3. Percentage of plant habits for the plant species of all selected sites around River Chenab, Punjab, Pakistan.**

**Table 2. Phytosociological analysis of 12 selected sites around River Chenab.**

| Sites | Soil type | Anthropogenic factors | Species documented | Perennials | Annual | Herbs | Shrubs | Trees |
|---|---|---|---|---|---|---|---|---|
| Site 1 Jammu Tavi, Bajwat | Loamy | Agricultural practices and grazing | 27 | 12 | 15 | *Marsilia quadrifolia, Azolla pinniata, Osmunda regalis, Cynodon dactylon* and *Ranunculus muricatus* | *Calotropis procera* and *Ipomoea carnea* | *Acacia nilotica* and *Broussonetia papyrifera* |
| Site 2 Head Maralla main bridge | Clay loamy | Picnic and parking | 18 | 7 | 11 | *Croton bonplandianus, Polygonum plebeium, Eleusine indica*, and *Nelumbo nucifera* | *Cannabis sativa* and *Lantana camara* | *Acacia nilotica* and *Eucalyptus globulus* |
| Site 3 Manawar Tavi Thokar Number 4 | Loamy | Flooding | 32 | 12 | 21 | *Eleusine indica, Cynodon dactylon*, and *Chrozophora tinctoria* | *Lantana camara, Adhatoda vasica* | *Acasia arabica* |
| Site 4 Canal Colony side | Loamy | Picnic place | 18 | 7 | 11 | *Tamarix aphylla, Asphodelus fistulosus* | *Ipomoea carnea* | *Morus nigra* |
| Site 5 Bella Forest area, Head Khanki | Loamy | Grazing and urbanization | 32 | 10 | 22 | *Leucas aspera, Digeria arvensis, Pontederia crassipes* and *Solanum nigrum* | *Ipomoea carnea* and *Datura alba* | *Eucalyptus globules* |
| Site 6 Head Khanki main bridge west side | Loamy | Construction | 26 | 18 | 8 | *Chrozophora tinctoria* | *Datura alba* | *Eucalyptus globulus* and *Ziziphus jujuba* |
| Site 7 Khanki village downstream | Loamy | Urbanization and road constructions | 20 | 12 | 8 | *Convolvulus arvensis, Osmunda regalis* and *Spirodela polyrhiza* | *Calotropis procera* and *Datura alba* | *Broussonetia papyrifera* and *Dalbergia sissoo* |
| Site 8 Lower Chenab Canal Khanki | Sandy loamy | Habitat fragmentation | 24 | 15 | 9 | *Eleusine indica, Polygonum plebeium* and *Azolla pinniata* | *Datura alba* | *Acacia nilotica* |
| Site 9 Chanowala wetland area Qadirabad | Loamy | Fishing and hunting of different birds | 20 | 13 | 7 | *Nelumbo nucifera* | *Ipomoea carnea* | *Accacia nilotica* |
| Site 10 Mureed large | Loamy | Waste material and sewage | 45 | 30 | 15 | *Nelumbo nucifera, Tribulus terrestris* and *Chrozophora tinctoria* | *Datura alba, Typha latifolia, Cannabis sativa* and *Cassia occidentalis* | *Morus nigra, Accasia arabica* and *Populas nigra* |
| Site 11 Mureed small | Loamy | Fish farming and fragmentation of natural habitat | 26 | 16 | 10 | *Cirsium arvensis* and *Lactuca sativa* | *Solanum xanthocarpum* and *Riccinus communis* | *Acacia nilotica* and *Ziziphus jujuba* |
| Site 12 Balloki-Link Canal | Loamy | Fish farming | 26 | 13 | 13 | *Silybum marianum* | *Parthenium hysterophorus* and *Lantana camara* | *Populas nigra* |

*Datura alba*) and 1 species of tree (*Eucalyptus globulus*). Further, phytosociological analysis of site 7 Khanki village downstream elaborated loamy soil which was highly disturbed by urbanization and road constructions. On this site 20 plant species were recorded in which 12 were annual and 8 as perennial. Dominant species of this site were herbs (*Convolvulus arvensis, Osmunda regalis* and *Spirodela polyrhiza*), shrubs (*Calotropis procera* and *Datura alba*) and trees (*Broussonetia papyrifera* and *Dalbergia sissoo*).

Phytosociological analysis of site 10 Mureed large; the soil condition of this site loamy and this site is mostly known as Bella area. Total 45 plant species were documented from this site in which 30 annuals and 15 species were perennial. The dominant plant species of this area were herbs including *Nelumbo nucifera, Tribulus terrestris* and *Chrozophora tinctoria* while in shrubs; *Datura alba, Typha latifolia, Cannabis sativa* and *Cassia occidentalis*. As trees, *Morus nigra, Acacia arabica* and *Populus nigra* were documented. Phytosociological analysis of site 11 Mureed small revealed 26 plant species in which 16 plants were annual and 10 perennial.

The diversified dominant plant species of this site were in herbs such as *Cirsium arvensis* and *Lactuca sativa*, while in shrubs *Solanum xanthocarpum*, *cannabis sativa* and *Riccinus communis*. Trees included as *Acacia nilotica* and *Ziziphus jujuba*.

Current studies also reported significant differences in alpha diversity of loamy soil and between loam and clay loam. These differences were confirmed with the Wilcoxon test. Different sites were exhibited varying diversity indices among different soil types. For example, site 3 and site 7 had clay loamy soil type and were significantly different from other sites. Loam, sandy and sandy loam soil types had significant values of alpha diversity. Shannon and Simpson alpha diversity indices were observed varying across all of the sites and soil types. Significant differences were also observed in alpha diversity between and within different soil types. Those sites which possess clay loam were different from each other which could be due to varied environmental factors (Fig 4).

Richness distributed species on the basis of individual plant species, it ranged from 8.88 to 22.25. It is pertinent to mention that the greater is the number of species, greater will be the value of richness. The values of Richness and Evenness are presented in Table 3. The correlation between the relative frequency and density showed the significant values (P ≤ 0.05). Likewise, the association between relative frequency and cover also showed a significant level (P ≤ 0.05) as compared to density and cover as non-significant (Table 4).

The Principal coordinate analysis (PCoA) described the variation of 120 plant species in 48 quadrats along different environmental factors in the form of plots (Fig 5). It was observed

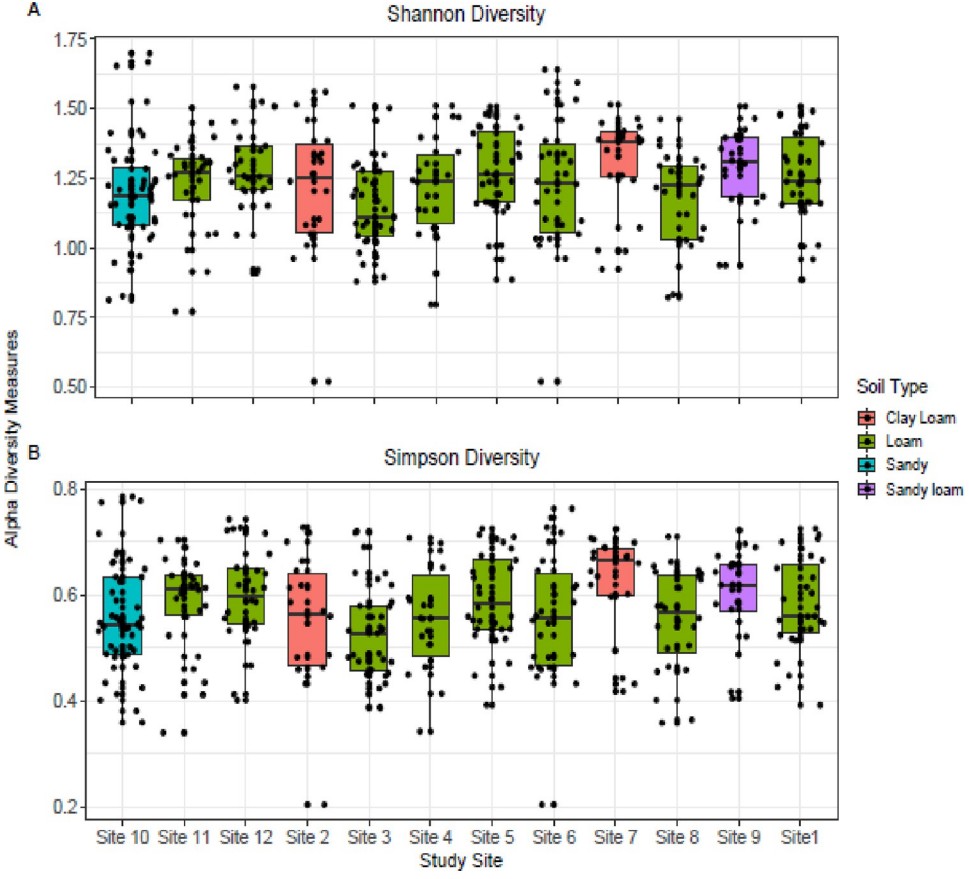

**Fig 4. Shannon and Simpson diversity indices for plant species of 12 selected sites around River Chenab, Punjab, Pakistan.**

**Table 3. Diversity indices, richness and evenness for the plant species of 12 selected sites around River Chenab.**

| S. No. | Site No. | Richness | Evenness |
|--------|----------|----------|----------|
| 1 | Site 1 | 14.18 | 0.96 |
| 2 | Site 2 | 8.88 | 0.88 |
| 3 | Site 3 | 15.56 | 0.97 |
| 4 | Site 4 | 9.90 | 0.95 |
| 5 | Site 5 | 15.19 | 0.91 |
| 6 | Site 6 | 13.69 | 0.95 |
| 7 | Site 7 | 10.07 | 0.93 |
| 8 | Site 8 | 12.83 | 0.92 |
| 9 | Site 9 | 10.68 | 0.93 |
| 10 | Site 10 | 22.25 | 0.95 |
| 11 | Site 11 | 12.92 | 0.96 |
| 12 | Site 12 | 12.79 | 0.94 |

that soil texture (loamy, clay loamy, clay and sand), organic matter, phosphorus, potassium, pH and electric conductivity (EC) showed significant effect on species distribution and composition. Loamy soil texture showed presence and dispersal of more vegetation as compared to loam, sandy and sandy loam soils.

Soil moisture contents were varied at different sites like at site 2 and 7 higher level of soil moisture was recorded as compared to site 9 and 10. These four sites were significantly varied for soil moisture at different sites (Fig 6). Additionally, available potassium was varied at different sites. Site 10 has significantly different values of available potassium which showed higher levels of plant diversity than other sites, hence available potassium exhibit a major effect on plant diversity (Fig 7). EC ds/m was varied in range for different sites. Site 4 has minimum value while site 12 with maximum value of EC ds/ m and these are significantly different from other sites because of linkage with soil saturation (Fig 8).

**SWOT analysis.** Results based on SWOT analysis deliberated that existence of vegetation and availability of food were the strength and an extreme importance of the study area. Medicinal importance, availability of natural resources and topography were at the moderate level and subtotal Importance Coefficient (IC) of the strength was 0.42 (Table 5). As far as opportunities were concerned, Forest and irrigation department were the main drivers to protect the wild flora, management strategies and the subtotal sum of IC were 0.22. As indicators of non-significant (negative) descriptors (weakness and threats), showed lower value which indicate worse negative impact. Further, weakness such as floods and agricultural practices were the extreme level and the subtotal sum of IC was 0.21. Similarly, threats, grazing and anthropogenic activities had extreme impact with the subtotal sum of 0.17.

**Table 4. Pearson correlation between relative frequency, density, and cover for the plant species of 12 selected sites around River Chenab.**

| | Relative frequency | Density |
|--------|---------------------|---------|
| Density | 0.96** | |
| | 0.00 | |
| Cover | 0.58* | 0.46* |
| | 0.05 | 0.13 |

P-Value 0.05%

**Highly significant

*High significant

**Table 5. SWOT analysis elaborating environmental factors/indicators, (scale range 100–1000) and importance coefficients (IC) used to determine habitat suitability of the riverine area of River Chenab.**

| SWOT group | SWOT factors/ Indicators | Descriptors | Weight | Importance coefficients (IC) |
|---|---|---|---|---|
| Internal origin | Strengths | Size of area | 100 | 0.05 |
| | | Food availability | 100 | 0.05 |
| | | Vegetation | 100 | 0.05 |
| | | Medicinal importance | 80 | 0.04 |
| | | Socioeconomic importance | 70 | 0.03 |
| | | connectivity | 70 | 0.03 |
| | | Climate | 80 | 0.04 |
| | | Topography | 70 | 0.03 |
| | | Available natural resources | 70 | 0.03 |
| | | Biodiversity | 80 | 0.04 |
| | | Ecological role of species | 30 | 0.01 |
| | | Subtotal | 85 | 0.42 |
| | Weaknesses | Risk of native species | 70 | 0.03 |
| | | Risk of diseases | 60 | 0.03 |
| | | Risk of flood | 90 | 0.04 |
| | | Soil erosion | 60 | 0.03 |
| | | Agricultural practices | 90 | 0.04 |
| | | Competition with other native species | 60 | 0.03 |
| | | Subtotal | 43 | 0.21 |
| External origin | Opportunities | Habitat protection by irrigation department | 80 | 0.04 |
| | | Management plan for the area | 70 | 0.03 |
| | | Forest Department | 80 | 0.04 |
| | | Existing conservation program | 50 | 0.02 |
| | | Conservation unit | 60 | 0.03 |
| | | Existing tourism involvement | 60 | 0.03 |
| | | Public involvement | 50 | 0.02 |
| | | Subtotal | 45 | 0.22 |
| | Threats | Competition by exotic plants | 60 | 0.03 |
| | | Grazing | 80 | 0.04 |
| | | Anthropogenic activities within the area | 60 | 0.03 |
| | | Risk of human infrastructure within area | 70 | 0.03 |
| | | Fish forming | 70 | 0.03 |
| | | Subtotal | 34 | 0.17 |
| | | Grand total | 21 | 10 |

**Strengths.** At site 1, Jammu Tavi, Bajwat; both terrestrial and aquatic ecosystems exits which have dense vegetation (Table 6). Most of the riverine area with soil texture as loamy, dominated by *Typha latifolia*, *Osmunda regalis*, *Calotropis procera* and *Broussonetia papyrifera*. This site was considered important for various socioeconomic purposes. For example, *Calotropis procera* was used for curing different ailments. *Typha latifolia* was helpful as making baskets, curtains and several domestic items (Fig 9B & 9C). This area is located in the foothills of Himalaya with varying climatic conditions reflecting diversified vegetation. Different birds like Cormorants, Indian Roller, Green Bee eaters and Pheasants were host in different seasons (Fig 9D).

At site 2, Main Head Maralla Bridge was designated for tourism and parking areas where several plant species present including *Leucas aspera*, *Phragmites karka*, *Convolvulus arvensis*

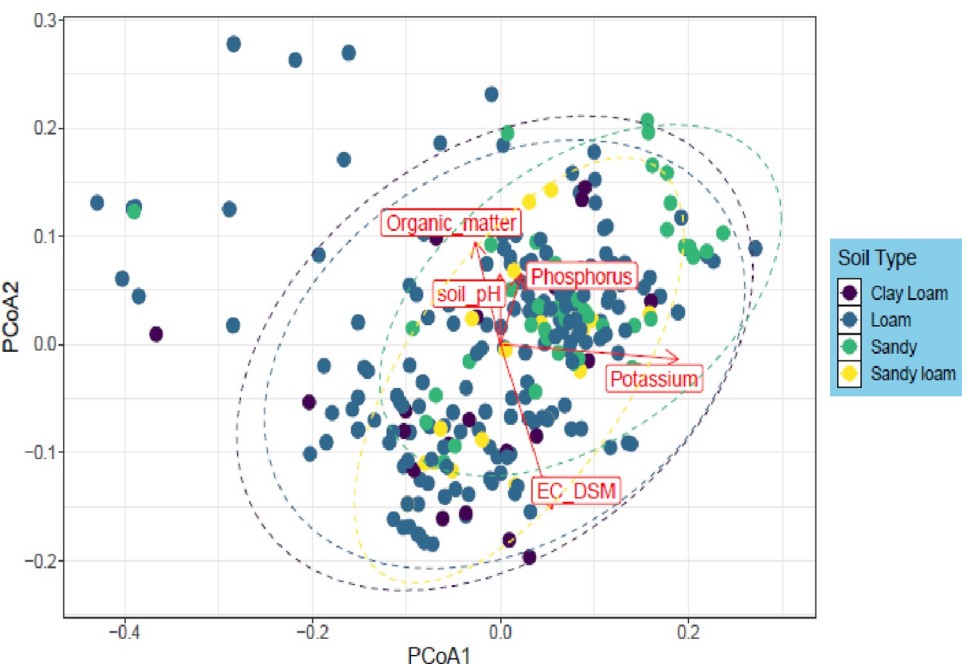

**Fig 5. Plant diversity with different soil types of 12 selected sites around River Chenab.**

and *Solanum xanthocarpum*. Out of theses, *Leucas aspera* and *Solanum xanthocarpum* generally preferred in the area as medicinal against digestive disorders. This site was located on the upstream of Head Maralla where all the water of river Chenab, river Jammu Tavi and Manawar Tavi regulated while catchment area was known as very fertile. Further, at site 3, Manawar tavi, Thokar Number 4, all three rivers viz., Manawar Tavi, Chenab and Jammu Tavi met and made a junction. The pH value of this site was 7.99 while organic matter as 0.72% (Fig 9A).

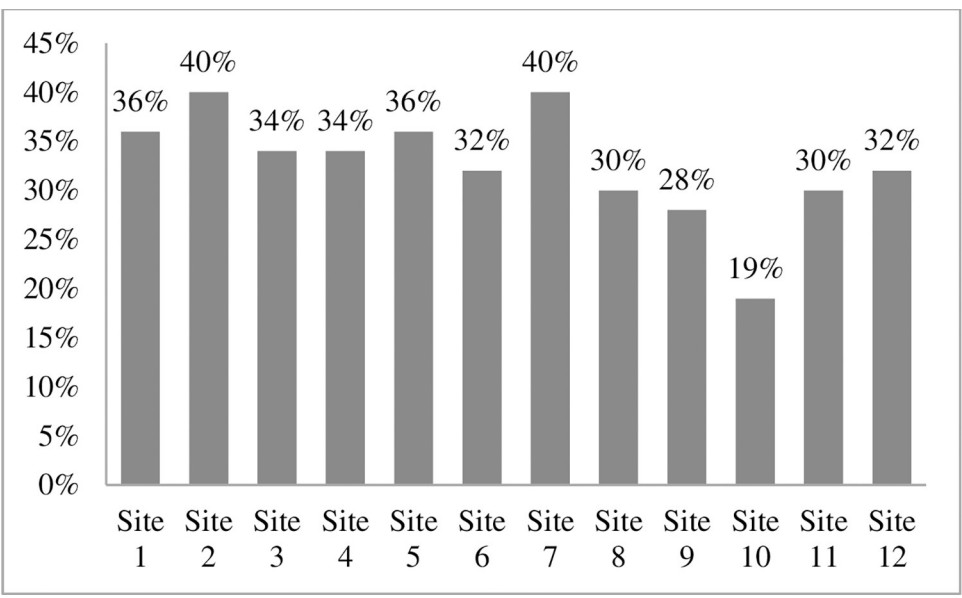

**Fig 6. Soil moisture (%) of 12 selected sites around River Chenab, Punjab, Pakistan.**

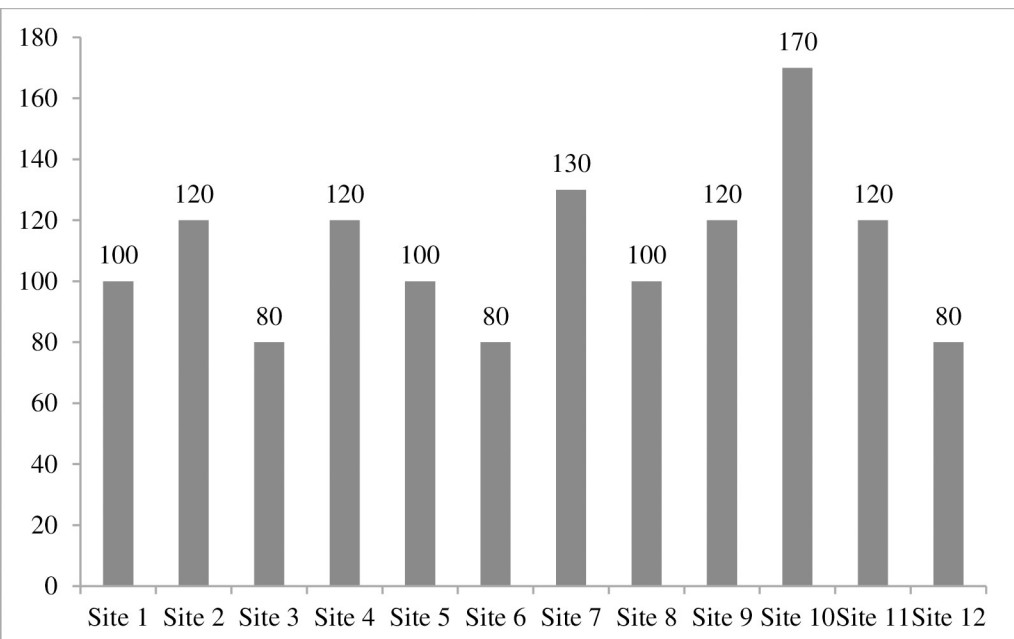

**Fig 7. Available potassium (ppm) of 12 selected sites around River Chenab, Punjab, Pakistan.**

At site 5, Bella Forest area, Head Khanki, upstream side was observed mostly the Bella area with riverine vegetation including *Datura alba*, *Croton bonplandianus*, *Ricinus communis* and *Populus nigra*. These plants were used for treating digestive disorders and circulatory diseases.

Furthermore, site 10, Mureed Large was located on the south bank of the river at Head Qadirabad. The soil texture of this site was sandy soil and the available potassium was 170 ppm. This site was found most diverse than the other sites. At site 11, Mureed small was located about 5 km south side of the head Qadirabad with loamy soil and organic matter was

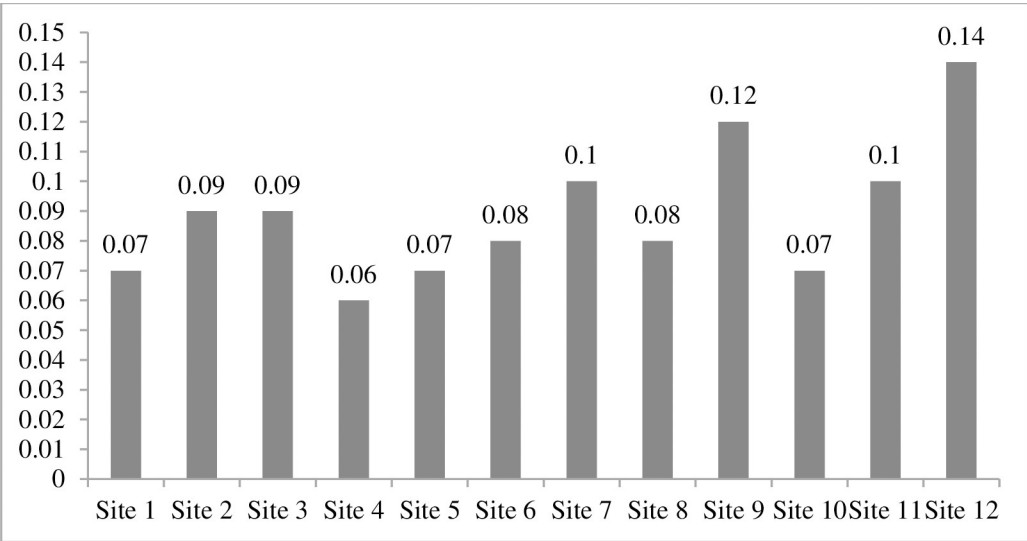

**Fig 8. EC (ds/m) of 12 selected sites around River Chenab, Punjab, Pakistan.**

**Table 6. Description of 12 sites studied with respect to SWOT analysis showing physical & chemical properties and dominant species across riverine area of River Chenab.**

| Site No. | Name of sites | Soil properties | Important species | SWOT factors/indicators | | | |
|---|---|---|---|---|---|---|---|
| | | | | Strengths | Weaknesses | Opportunities | Threats |
| Site 1 | Jammu Tavi, Bajwat area | Loamy soil with pH 8.13 and organic matter 0.68 | *Typha latifolia, Osmunda regalis, Calotropis procera* and *Broussonetia papyrifera* | Place for various socioeconomic activities. *C. procera* used as medicinal while *T. latifolia* as basket making, curtains and several domestic items. Host of several birds | Overuse of medicinal plants, habitat destruction, land conversion for agricultural purposes and the catchment areas used as grazing | Tree plantation by Forest Department to stop soil erosion and promote conservation | Flooding |
| Site 2 | Head Maralla main Bridge | Sand loamy soil with pH 8.17 and available potassium 120 ppm | *Leucas aspera, Phragmites karka, Convolvulus arvensis* and *Solanum xanthocarpum* | Tourism and parking, *L. aspera* and *S. xanthocarpum* used as medicinal against digestive disorders. Catchment area was considered as very fertile. | Waste damping especially food stuff | Plantation by Punjab Irrigation Department | Flooding and parking areas |
| Site 3 | Manawar Tavi, Thokar Number 4, junction of three rivers | pH 7.99 and organic matter 0.72% | *Lantana camara, Adhatoda vasica* and *Acacia arabica* | *A. vasica* mainly used as medicinal for treating several disorders | Over harvesting and over exploitation of natural biodiversity | Forest Department planted different trees and artificial forests for conservation | Flooding and grazing |
| Site 4 | Canal colony side located at the downstream of Head Maralla | pH 8.45 and available phosphorus 5.4 ppm | *Tamarix aphylla, Asphodelus fistulosus* and *Ipomoea carnea* | Tourism and picnic | Different anthropogenic activities involving tourism and parking areas | Tree plantation in canal colony sides and the catchment area is restricted for any general activity | urbanization by transferring agricultural land for housing purposes |
| Site 5 | Bella Forest area, Head Khanki, upstream side | pH 8.42 and organic matter 0.77% | *Datura alba, Croton bonplandianus, Ricinus communis* and *Populus nigra* | Documented plants were being used for treating digestive disorders and circulatory diseases | Animal grazing, flooding and soil erosion | Forest Department planted different reserve forests | Habitat fragmentation |
| Site 6 | Head Khanki main Bridge downstream | Loamy soil with 8.5 pH and 0.77% organic matter | *Silybum marianum, Chrozophora tinctoria, Cassia occidentalis* and *Ziziphus jujuba* | *Chrozophora tinctoria* is used for veterinary treatments | Construction of a bridge and Human implementations like urbanization | Local people contributed themself for the maintenance of wild areas | Construction of the Khanki Bridge |
| Site 7 | Khanki village located at the downstream of head Khanki | Clay loamy with 40% soil moisture | *Solanum nigrum, Typha latifolia* and *Dalbergia sissoo,* | *T. latifolia* for making of domestic goods and *D. sissoo* for furniture making. *S. nigrum* used for the treatment of the digestive disorders | Construction of main Head Khanki Bridge and several anthropogenic activities such as habitat fragmentation | Forest department contributed plantation for rehabilitation of the Head Khanki area | Use of heavy construction machinery |
| Site 8 | Lower Chenab Canal of Head Khanki | Loamy soil with pH 8.43 and 0.64% organic matter | *Polygonum plebeium, Capsella bursa pastoris, Ipomoea carnea* and *Eucalyptus globules* | Plants of the area mainly known to cure various maladies | Agricultural practices such as over land use and lower connectivity level due to the construction of Head Khanki Bridge | Irrigation department plantation helps in conservation of natural vegetation | Habitat fragmentation |
| Site 9 | Chanowala wetland area Qadirabad | Sandy loamy with EC 0.12 ds/m and 0.77% organic matter | *Nelumbo nucifera, Pistia stratiotes, Eichhornia crassipes* and *Ipomoea carnea* | Chanowala wetland area Qadirabad, considered as diverse area for aquatic vegetation | Disturbed by human implementations and land use for multiple purposes | Wildlife department and forest department Mandi Bahudin conserving area | Wetland areas are at lower altitude than the river, flooding, fishing and hunting |
| Site 10 | Mureed Large, South bank of the river at Head Qadirabad | sandy soil and available potassium 170 ppm | *Tribulus terrestris Chrozophora tinctoria* and *Typha latifolia* | Diversified vegetation site as compared to other sites | Unorganized agricultural practices were the main challenge for this site | Forest department plantation of different artificial and reserved forests for the conservation | Polluted area by waste material and sewage |

(*Continued*)

**Table 6.** (Continued)

| Site No. | Name of sites | Soil properties | Important species | SWOT factors/indicators | | | |
|---|---|---|---|---|---|---|---|
| | | | | Strengths | Weaknesses | Opportunities | Threats |
| Site 11 | Mureed small, 5 km south side of the head Qadirabad | Loamy soil and organic matter 0.55% | *Nasturtium officinalle, Inula hirta, lamium amplexicaule* and *Alhagi maurorum* | *A. mourorum* used for cough, fewer, constipation and skin diseases. | Fish farming | Forest department plantation | habitat loss and fragmentation |
| Site 12 | Balloki Link Canal, located on Head Qadirabad | Loamy soil with available phosphorus 6.7 ppm | *Physalis minima, Potentilla argentea, Saccharum spontaneum* and *Ricinus communis* | *P. minima* used for digestive disorders and *R. communis* leaves used for pain killer and reducing snake poisonous. | habitat destruction and fish forming | Irrigation and Forest Department plantation | Urbanization |

0.55%. Main vegetation of this site was *Nasturtium officinale, Inula hirta lamium amplexicaule* and *Alhagi maurorum*.

**Weaknesses.** At site 1, Jammu Tavi, Bajwat area, by demographically people depend on plant biodiversity as the source of income. Number of anthropogenic activities were noted in the area as over exploitation of medicinal plants, habitat destruction, conversion of land for agricultural purposes and the catchment areas using for grazing (Fig 9F). At site 2, Main Head Maralla Bridge, where people used land for tourism and parking. Another weakness of the area is the damping of waste material especially food at the same point. At site 5, Bella Forest area; animal grazing and flooding were the main causes for disruption of the vegetation. On the other side, in lower catchments, soil erosion formed small ditches and lower water bodies disturbed the vegetation as shown in Fig 9E. At site 6, Head Khanki west side; natural vegetation of the site had been disturbed by construction of a new bridge which was the one of the main factor for habitat destruction (Fig 9G). Site 9, was once considered a diverse area for aquatic vegetation but now disturbed by human intervention and land use for multiple purposes as well site 10 faced the challenges of access agricultural practices, While site 11 hampered by fish farming and 12 by both habitat destruction and fish farming.

**Opportunities.** The forest department, Punjab irrigation department and Pakistan army protect the vegetation and planted various tree species in the catchment area of river Chenab to stop soil erosion and promote conservation at various dites At site 5, the forest department planted different reserve forests for the conservation of the environment whereas locals were restricted to observe the maintenance of wild areas along site 6. At site 7, the forest department has major contribution for the rehabilitation of the Head Khanki area. Likewise, at site 9, Wildlife department and forest department Mandi Bahudin protecting the aquatic ecosystem and conserving the wild flora and fauna. All other sites were also maintained by various departments.

**Threats.** At site 1, the main threat was flooding due to Jammu Tavi River which has no marker bank in the catchments. In the monsoon season, when the water table rises all surrounding areas become flooding. Site 3, Junction of three rivers, has been observed with high rate of flooding at this site. Local inhabitants from this site graze their livestock that was another threat of this site. Similarly, site 4 posed pressure of anthropogenic activities mainly urbanization while site 5 inclined towards habitat fragmentation and grazing. Soil erosion can be seen at various places which was caused by rise in water table due to flooding. Furthermore, the construction of Head Khanki Bridge caused habitat destruction at site 6.

At site 8, habitat fragmentation was the main threat because of waste material related to heavy machinery that badly affect floral diversity (Fig 9H). In the same context, at site 10, was also prominent where waste material and sewage were main threats polluting the riverine area (Fig 9I).

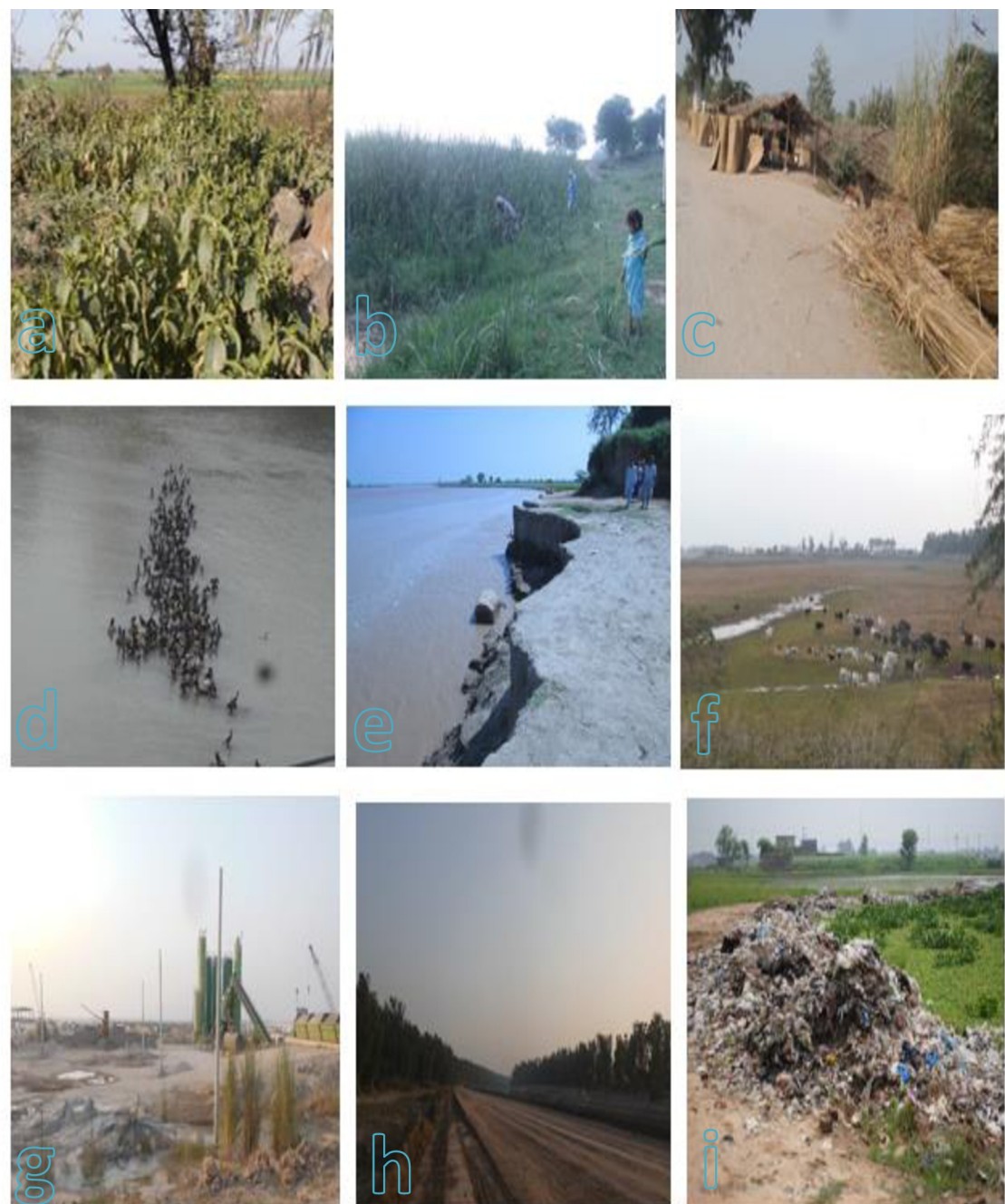

**Fig 9.** Selected pictures of the area around River Chenab studied for SWOT analysis [a. *Adhatoda vasica*, widely available medicinal plant of the study area, b. Collection of *Typha latifolia*, c. Uses of *Typha latifolia* for different purposes, d. Birds at wetland area site, e. Soil erosion, f. Grazing, g. Urbanization, h. Habitat fragmentation, i. Waste material at the catchment area].

## Discussion

For comprehensive elaboration of the area with respect to phytosociology and factors influencing vegetation, present studies can be divided into three parts; first part describes the floral diversity and phytosociological analysis of the vegetation in the catchment area of river Chenab from Head Maralla to Head Qadirabad with 12 sites. The second part represents the

edaphic factors such as physical and chemical properties of the soil because it played key role in species distribution and diversity. In the third part, SWOT analysis revealed description of both internal and external factors. Such type of studies provide information about the interaction between different biotic and abiotic components of an ecosystem [10], while phytosociological attributes help to understand change in vegetation structure in heterogenic environment [35].

## Floristic diversity

Checklists for floristic diversity are the most useful resource for botanical information of a particular area. It is a useful marker because it demonstrate variation in different environments with varying environmental factors that lead to inter and intraspecific diversity [36]. Thereby, indigenous flora and its identification are very important for introducing particular species of the specific area with their characteristics under climatic conditions like temperature and drought [37, 38].

Fifty one plant families were studied in the catchment area of river Chenab started from Head Maralla, Sialkot to Head Qadirabad, MandiBahudin during summer 2019. Documented families showed several traits in common along variation with vegetation reported from other parts of the country, e.g., Asteraceae and Poaceae were the most dominant families [39, 40].

## Phytosociological analysis

The study area river Chenab and its catchment area contain mostly wetland area, wild area, reserve forest and water bodies like canals, ponds, ditches and drains from which 12 sites were selected for sampling and phytosociological studies conducted for the quantitative attributes. In the past, several phytosociological studies about indigenous flora from various parts of the country have been reported by many researchers [10, 41–46]. In the present studies, 120 plant species and 51 families were recorded in which dicot families were 40 and 3 monocots. The dominant families were Asteraceae, Fabaceae and poaceae. Current findings are congruent with those of which reported the poaceae family was the dominant family of this site and several others were similar throughout several areas [47–50]. From the subject area, 89 herbs, 16 shrubs and 15 tree species were recorded. In perennial plants mostly members of Fabaceae and Moraceae family were dominant. Further, the recorded value lies within the reported range for other areas of Pakistan [51].

In our results, site 3 has highest value of evenness 0.971 while site 2 has lowest value with 0.889, rest of the sites has moderate values. In Evenness, the Shannon index is divided by the natural log of the number of species [52, 53]. In richness, site 10 has maximum value with 22.25 that shows the species diversity of the site. The species richness interpreted the biological variation due to various environmental factors hence both in space and time, abiotic factors affect vegetation structure [5, 54]. Shannon index of site 3 and 7 have different results than other sites that depend on the soil type. These findings also open new ways to study and manage phytoecological services and environmental sustainability. The correlation between relative frequency and density were highly significant than other parameters. Current findings, based on the ecological approaches, can be used as one of the criteria for prioritization of special habitats, protected areas and for conservation of species on strategic basis across river Chenab. From our findings *Adhatoda vasica*, *Geranium rotundifolium*, *Argemone Mexicana* and *Thymelia passive* were unique species which were not common on other sites. Additionally, in soil type, Loamy soil and sandy loamy soil have significant role in species diversity. As any change in environmental factors and altitude cause significant impact on formation of plant species [55, 56].

Site 10 has loamy soil and has maximum diversity. Further, soil is an environmental factor that determines plant growth which is influenced by organisms, topography, climate, parent material and time [57]. By origin various plant species restricted to the specific habitat or adopted to settle in the same vicinity due to the presence of optimum environmental factors including soil chemistry which clearly show that vegetation composition changes along the ecological diversity from place to place. There is 85 km distance from site 1 Jammu Tavi, Bajwat area, upstream side of Head Maralla to site 12 Balloki Link Canal, Head Qadirabad which comprehends with varied vegetation from site to site. Site Number 2 and 4 exhibited minimum plant species as compared to other sites. Plant diversity was being disturbed by using land as parking area and construction of commercial purpose buildings at site 2 while urbanization by means of housing colonies lowers biodiversity values.

Similarly, low species diversity was also reported from Kotli area which was because of deforestation, human interventions, exploitation of medicinal plants and cold condition that causes quick disappearance of annual plants. This represents as an environmental stress [58]. Current findings also expressed concern that species diversity was low due to a smaller number of species at lower altitude and high due to the high number of species at high altitude. It is therefore suggested that conservation of rare species may accompanied with special or fragmented habitats. Moreover, it is recommended to categorize species under IUCN categories to take an effective action for timely and systematic conservation. Site 10 exhibited more plant species than any other site that could be attributed with low anthropogenic activities and area surrounding the river remaining as wild core area. Soil analysis showed that available potassium had maximum value than other sites which played significant role in alpha diversity. Supporting results for plant species composition were also reported from other regions of the country like Dao Khan Hills, Azad Jammu & Kashmir and Central Himalayan (Kumaun) region of India [50, 59].

Additionally, alpha diversity across different sites varied with environmental factors. Site 3 and 7 have clay loamy soil and were non-significant than other sites. Alpha diversity mainly depends on the soil structure that varies from site to site. Other sites showed the significant results of alpha diversity, similar pattern of diversity across environmental factors have been observed in other studies in the different regions of the country like Naran Valley [60–62]. Both physical and chemical parameters of soil are important for alpha diversity. Greater the number of individuals greater will be the value of richness. Loamy soil has significant role in alpha diversity because it showed maximum plant diversity as compared to other soil structures. Alpha diversity indices across different sites based on soil types explain that vegetation pattern varies with different soil types. Similarly, sites existed on the same soil type have low diversity which could be supported by the effects of other abiotic or biotic factors [7].

Site 10 possessed highest value of richness than other sites while site 3 had highest value of evenness [5, 58, 61, 63]. Dominant tree species recorded from the site were including *Acacia nilotica* L., *Acacia arabica* Lam., *Broussonetia papyrifera* L., *Dalbergia sissoo* Roxb. and *Ziziphus jujube* Mill., which were also reported in the Naran valley of the western Himalayas [62].

## SWOT analysis

SWOT analysis approached was adapted to characterize the positive and negative aspects for conservation planning for riverine area of river Chenab. Findings of the SWOT analysis can be used to develop specific management strategies to increase and maintain suitability of the selected sites. It was evident that area quality indices vary from descriptors to descriptors along comprehensive internal and external factors [32]. For indicators of positive descriptors (Strength and opportunities) and higher values indicate greater positive impact [64, 65]. For

current findings, quantitative SWOT analysis has desirable characteristics as applied on riverine area. This method is versatile, repeatable, intuitive, help to identify and reduce uncertainties, additionally highlight topic scenarios [66, 67].

The SWOT analysis comprehensively provided both internal and external factors (strengths and opportunities) and vulnerabilities (weaknesses and threats) faced by vegetation across river Chenab. As SWOT table describes several anthropogenic activities affected the overall biodiversity, resultantly a negative impact on surrounding vegetation hence, depleting natural resources. The outcomes provide helpful information for multiple scale perspective, i.e. negative and positive aspects at local, regional or national level. In terms of qualities (strengths and opportunities), an ideal site would be located on the head Maralla wetland area where anthropogenic activities are minimal as compared to other wetland areas. At site 1 Jammu Tavi Bajwat area, located near the Bajwat wildlife sanctuary, declared as game reserve area where human implementations were minimum and high biodiversity still intact than other sites because this site located in the foothill of Himalaya region. Departments like Forest Department and Irrigation planted trees and protected the natural flora in the catchment area of river Chenab.

In weakness, soil erosion and grazing are the main negative aspects. For other wetlands, anthropogenic activities such as urbanization and habitat fragmentation were the major threats [68]. Loss of habitat can break the connectivity between the ecological linkages with other surrounding areas [69].

By using the quantitative method, it shows that plants were being used for different purposes and socioeconomic importance of the area. Head Maralla has a larger core area with fewer disturbances than other sites. Areas with fewer disturbances have more bestowed with the natural assets [70]. On the other hand, at the head Khanki wetland area, major threat was the habitat fragmentation and urbanization due to construction of bridges. While at head Qadirabad wetland area, vegetation was disturbed by the habitat fragmentation in aquatic land. Building social and political support and involvement of local communities is an important component which can mitigate the human implementation that could be helpful for the conservation of plant biodiversity [71]. Quantitative description of plant diversity in current findings is characteristic and can be used for confirmation, comparison and assessment of anthropogenic pressures while planning conservation strategies. Documentation which critically evaluates plant biodiversity and the factors driving it at national and regional levels are mandated by law in the developed world. This approach can be adopted in the developing world as well for long-term management and sustainability of the natural resources [72]. This study comprehensively provides baseline data/information to assess various environmental effects on plant species distribution and to devise conservation management for the future.

## Future prospects

Observing and studying plant biodiversity is a lifelong process and needs input from many discipline including natural as well as social sciences. For better understanding, plant biodiversity should be studied at species, community and ecosystem levels. The catchment areas of river Chenab need more botanical exploration for the complete description of their plant taxa, richness and conservation status. There is huge gap in information on assessment and monitoring of plant biodiversity of river Chenab. Most of the report on vegetation have been carried out exclusively either based on scientific approaches for making inventories. Therefore, there is need to understand both environmental and SWOT factors together because this is the right time to examine species diversity in the context of climate change and green house effects. Rising temperature and deforestation by means of anthropogenic practices lining unrepairable

losses to plant biodiversity. Hence, focus should be towards considering all possible drivers of plant biodiversity across areas and regions. Current findings suggests that SWOT as an approach can be adapted from the individual level to whole plant community level for better understanding of present plant biodiversity status and future threats. Furthermore, it is required to communicate to conservationists, policy makers, and planners. Starting from site 1 to another will be next step to take first concrete action in the light of the findings with the available resources.

Few of the unique species can play significant role in monitoring and assessment of the biological diversity on the one hand and positive and negative aspects on other. To overcome emerging challenges steps required to be taken not only on root level as local community level but as a regional policy through strict monitoring policies, which always requires willingness of all stack holders. Consequently, role of local government with community groups and NGOs is imperative to stop broad scale deterioration of the natural ecosystems due to increase in population, agriculture, deforestation and expansion of roads. It is not only the responsibility of the riverine community but of all for all to implement conservation policies for the sustainable use of the plant biodiversity for coming generation.

## Future conservation

Existence of plant biodiversity along the river Chenab is not only hosting present day food, fodder, fuel, timber and medicinal plants but also to the wild relatives of cultivated plants which may possess useful stress and disease resistant characteristics such as *Desmostachya bipinnata* L., *Poa annua* L., *Saccharum arundinaceum* Retz., and *Saccharum spontaneum* L. Simply these are wild germplasm or genetic resources. Moreover, there are species which are not important economically but have a significant ecological role in controlling flood and soil erosion as by *Adhatoda vasica* L. Conservation of biodiversity and their sustainable use in the scenario of climatic change can only be possible if the important species and habitats are carefully managed, examined and preserved like *Bombax ceiba* L. Summarizing the overall description for future management of the river Chenab two broad strategic approaches could be implemented to preserve ecological diversity. Firstly, stop illegal consumption of natural resources and secondly, restoration of plant biodiversity through reforestation, designating protected areas and multiplying rare species locally. Another point is provision of basic health care facilities to the local people to conserve over exploitation of medicinal plants. This study could be a baseline source of data for the management and conservation planning of the river Chenab and its catchment area. The solution to the problems is also addressed through following Biodiversity action plan of Pakistan too and struggle for achieving sustainable development goals [73].

Present studies recommends designation of new protected areas like riverine area of river Chenab and Bella Forest. Further, launching nurseries would also be a timely contribution to protect local flora and in broader spectrum, establishment of satellite seed and/or gene bank could also help in achieving conservation goals.

## Conclusion

In dicot families, asteraceae family was the most abundant while in monocot poaceae present in wide range. Site 1 have 3 species of pteridophytes, 1 of bryophytes while at site 3 unique species *Adhatoda vasica* L. was common. Due to variation in soil texture, pH and potassium availability, site 3 and 7 were notably distinct from other sites in term of alpha diversity and soil type. Site 10 was found as most abundant having 45 plant species with greater value of richness 22.25. In evenness, site 3 has maximum value 0.971. The relative frequency and density showed

high correlation than other parameters. Based on soil chemical properties, site 2 and 4 shows the greater value of soil moisture content. SWOT analysis demonstrate that as strengths at the Head Maralla site, peoples were using plants for medicine, food and other economic purposes. In weakness, many internal factors like agricultural practices, soil erosion and flooding effected the vegetation, therefore need is to work towards such deficiencies. In opportunities, at Head Maralla, Head Khanki, Head Qadirabad, Forest and Irrigation Departments were planting different plant species for the restoration of ecosystem along the river. In threats, anthropogenic threats on different wetland areas were mainly overgrazing and urbanization including Head Maralla, habitat fragmentation at head Khanki, and extensive fish farming at head Qadirabad. The study strongly suggests immediate conservation of the areas as strategically rather ignoring before adverse consequences. Need is to take solid steps including awareness among local communities about possible impact of destroying natural ecosystem as evident from current climate change impact in the shape of intense heat wave and water shortage, hence sustainable utilization of natural resources by implementing Biodiversity Action Plan of Pakistan is recommended.

## Supporting information

**S1 Table. Checklist of floral diversity in order to botanical names, families, habits and groups from the study area.**
(DOCX)

## Acknowledgments

The authors highly acknowledge the assistance by Dr. Danish Ibrar, SSO, Oilseeds Research Program and Dr. Shahbaz Khan, SSO, Land Resources Research Institute, NARC, Islamabad for their inputs for improving the manuscript language. Authors are thankful to Dr. Khalid Iqbal, Office of Research, Innovation and Commercialization (ORIC), University of Gujrat, Gujrat, for grammer check, language editing and proof reading.

## Author Contributions

**Conceptualization:** Muhammad Azhar Ali, Muhammad Sajjad Iqbal.

**Data curation:** Muhammad Azhar Ali, Syed Atiq Hussain, Noshia Arshad, Saba Munir, Hajra Masood.

**Formal analysis:** Muhammad Akbar.

**Project administration:** Muhammad Sajjad Iqbal.

**Resources:** Ansar Mehmood, Tahira Ahmad.

**Software:** Khawaja Shafique Ahmad, Tahira Ahmad.

**Writing – original draft:** Muhammad Azhar Ali, Muhammad Sajjad Iqbal.

**Writing – review & editing:** Ghulam Muhiyuddin Kaloi, Muhammad Islam.

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
