## [Decision Letter · Decision Letter 0]

24 Jan 2022

PONE-D-21-39895Plant species diversity assessment and monitoring in catchment areas of river Chenab, Punjab, PakistanPLOS ONE

Dear Dr. Iqbal

Thank you for submitting your manuscript to PLOS ONE. After careful consideration, we feel that it has merit but does not fully meet PLOS ONE’s publication criteria as it currently stands. Therefore, we invite you to submit a revised version of the manuscript that addresses the points raised during the review process.

Please submit your revised manuscript by Mar. 10 2022. If you will need more time than this to complete your revisions, please reply to this message or contact the journal office at plosone@plos.org. Please include the following items when submitting your revised manuscript:A rebuttal letter that responds to each point raised by the academic editor and reviewer(s). You should upload this letter as a separate file labeled 'Response to Reviewers'.A marked-up copy of your manuscript that highlights changes made to the original version. You should upload this as a separate file labeled 'Revised Manuscript with Track Changes'.An unmarked version of your revised paper without tracked changes. You should upload this as a separate file labeled 'Manuscript'.

We look forward to receiving your revised manuscript.

Kind regards,

Tunira Bhadauria, Ph.D.

Academic Editor

PLOS ONE

Journal Requirements:

3. We note that Figure 1 in your submission contain map images which may be copyrighted. All PLOS content is published under the Creative Commons Attribution License (CC BY 4.0), which means that the manuscript, images, and Supporting Information files will be freely available online, and any third party is permitted to access, download, copy, distribute, and use these materials in any way, even commercially, with proper attribution. For these reasons, we cannot publish previously copyrighted maps or satellite images created using proprietary data, such as Google software (Google Maps, Street View, and Earth). For more information, see our copyright guidelines: http://journals.plos.org/plosone/s/licenses-and-copyright.

a) You may seek permission from the original copyright holder of Figure(s) [#] to publish the content specifically under the CC BY 4.0 license.  

Additional Editor Comments:

The document is significant in terms of diversity and conservation. As a result, the research was successful. It still needs a few minor tweaks.

1. The entire MS is written in a highly descriptive style, as if it were a book chapter, whereas it is always recommended that the MS be written in a crisp manner. As a result, authors should concentrate on this.

2. Similarly, the fraction of the result devoted to phytosociological analysis is excessively enormous.

3. The MS is tough to grasp since some lines are not nicely organised. For instance, lines 54-56, 59-60, 227-230, and 241-243.

4. Table 2 is optional in the main MS; however, it can be included as a supplementary table.

5. Go over the manuscript for grammar and spelling errors.

Reviewers' comments:

Reviewer's Responses to Questions

**Comments to the Author**

1. Is the manuscript technically sound, and do the data support the conclusions?

Reviewer #1: Yes

Reviewer #2: Yes

2. Has the statistical analysis been performed appropriately and rigorously? 

Reviewer #1: No

Reviewer #2: Yes

3. Have the authors made all data underlying the findings in their manuscript fully available?

Reviewer #1: Yes

Reviewer #2: Yes

4. Is the manuscript presented in an intelligible fashion and written in standard English?

PLOS ONE does not copyedit accepted manuscripts, so the language in submitting articles must be clear, correct, and unambiguous. Any typographical or grammatical errors should be corrected at revision, so please note any specific errors here.

Reviewer #1: No

Reviewer #2: Yes

5. Review Comments to the Author

Reviewer #1: The manuscript is of importance with respect to Diversity and its conservation. Thats why the study carried out is good. Still, it needs few minor corrections.

1. The whole MS is written in very descriptive manner like a book chapter whereas it is always recommendable to write the MS in crispy fashion. Therefore, authors are advised to focus on this.

2. Likewise, the result portion having phytosociological analysis part is too large.

3. Some lines in the MS are not well organised therefore it is difficult to understand. For example line number 54-56, 59-60,227-230, 241-243.

4. The table 2 is not required in the main MS, it may be added as a supplementary table.

5. Check the grammar and spelling error throughout the manuscript.

Reviewer #2: manuscript may be accepted after following suggested improvement

1.Pl explain future prospective

2.Pl improve english grammer

3.Line no 632-638,Pl explain how anthoropogenic activities are affecting biodiversity and what would be possible effort to improve It?

4.Pl elaborate discussion

5.Discussion is very poor pl improve it and corelate it with your data

6.Ethics statement must be at end of the manuscript or do needfull as per guideline of PLOS ONE

7.Author are suggested to revisit and remove repeating of centences .

6. PLOS authors have the option to publish the peer review history of their article (what does this mean?). If published, this will include your full peer review and any attached files.

Reviewer #1: No

Reviewer #2: No

---

## [Author Response · Author response to Decision Letter 0]

11 Jul 2022

Response to Reviewers comments [PONE-D-21-39895] - 

Reviewer #1: The manuscript is of importance with respect to Diversity and its conservation. Thaws why the study carried out is good. Still, it needs few minor corrections.

1. The whole MS is written in very descriptive manner like a book chapter whereas it is always recommendable to write the MS in crispy fashion. Therefore, authors are advised to focus on this.

We have improved the MS sufficiently to meet international standard of readership and made the MS crispier as recommended to avoid unnecessary length in sentences and meaning of the words. Table 2 and 6 have been prepared to reduce the excess of data for phytosociological analysis and SWOT in results section. 

2. Likewise, the result portion having phytosociological analysis part is too large.

Extra material have been removed.

3. Some lines in the MS are not well organized therefore it is difficult to understand. For example line number 54-56, 59-60,227-230, 241-243.

Following have been addressed;

Line 54-56

The best simply interpretable indication of biological variation is species richness which highlight by different environmental variables

In biological variation, the best interpretable indication is species richness which is highlighted by various environmental factors such as temperature, annual rainfall and soil composition.

Line 59-60

Scientists study numerous factors (abiotic and biotic) which interact in community composition formation.

Scientist studied both biotic (producers and consumers) and abiotic (Soil, temperature, pH) factors which are interrelated in community composition and formation.

Line 227-230

The particular number of SWOT indicates may change with sites. Indicators are allocate weight ranges from 100 to 1000 in term of relative importance (100 lower value of importance 1000 higher value of importance) of the factors in the study area.

SWOT values have particular number within the specific site. These values vary from 100 to 1000 in term of relative importance of weight ranges. 100 indicate the lower value of importance and 1000 indicate the higher value of importance of the factors in the study area.

Line 241-243

Different diversity indices such as Shannon- Weiner and Simpson index and also evenness and richness were calculated for the correct species diversity.

For the correct species diversity assessment, various diversity indices including Shannon-Weiner, Simpson, evenness and richness were studied and calculated.

4. The table 2 is not required in the main MS, it may be added as a supplementary table.

Shifted as supplementary Table ‘S1’ 

5. Check the grammar and spelling error throughout the manuscript.

Counter checked with ‘The plant list’ and English spelling mistakes through ‘Grammerly’ software

Reviewer #2: manuscript may be accepted after following suggested improvement

1.Pl explain future prospective

Following material has been included as Future Prospects in the main text

Future Prospects

Observing and studying plant biodiversity is a lifelong process and needs input from many discipline including natural as well as social sciences. For better understanding, plant biodiversity should be studied at species, community and ecosystem levels. The catchment areas of river Chenab need more botanical exploration for the complete description of their plant taxa, richness and conservation status. There is huge gap in information on assessment and monitoring of plant biodiversity of river Chenab. Most of the report on vegetation have been carried out exclusively either based on scientific approaches for making inventories. Therefore, there is need to understand both environmental and SWOT factors together because this is the right time to examine species diversity in the context of climate change and green house effects. Rising temperature and deforestation by means of anthropogenic practices lining unrepairable losses to plant biodiversity. Hence, focus should be towards considering all possible drivers of plant biodiversity across areas and regions. Current findings suggests that SWOT as an approach can be adapted from the individual level to whole plant community level for better understanding of present plant biodiversity status and future threats. Furthermore, it is required to communicate to conservationists, policy makers, and planners. Starting from site 1 to another will be next step to take first concrete action in the light of the findings with the available resources.

Few of the unique species can play significant role in monitoring and assessment of the biological diversity on the one hand and positive and negative aspects on other. To overcome emerging challenges steps required to be taken not only on root level as local community level but as a regional policy through strict monitoring policies, which always requires willingness of all stakeholders. Consequently, role of local government with community groups and NGOs is imperative to stop broad scale deterioration of the natural ecosystems due to increase in population, agriculture, deforestation and expansion of roads. It is not only the responsibility of the riverine community but of all for all to implement conservation policies for the sustainable use of the plant biodiversity for coming generation. 

2.Pl improve english grammer

Improved to our best and we acknowledge the services of native speakers of English Dr. Danish Ibrar and Dr. Shahbaz Khan to whom names are being acknowledged.

3.Line no 632-638,Pl explain how anthoropogenic activities are affecting biodiversity and what would be possible effort to improve It?

The present research describes the distribution of plant species along the river Chenab with relation to the environment. Different sites possess wide range of soil texture, pH and potassium availability which have a significant impact on plant species diversity. Site 3 and 7 are notably distinct from other sites in term of alpha diversity and soil type. Moreover, the density of species and relative frequencies were substantially associated with each other. Different anthropogenic activities effect the vegetation along the riverine area of river Chenab as at Head Maralla, over grazing were the main threat where farmers of the surrounding areas grazed their cattle’s at large scale. At head Khanki, several activities including mainly construction area, urbanization and habitat fragmentation disturbed the wild area. Furthermore, at head Qadirabad, fish farming was the main threat. Local community along the river converting and using their land for fish farming at alarming scale therefore, vegetation has badly affected. 

As remedy to conserve the flora of the area several conservation strategies like awareness among local community about the facts of destroying natural ecosystem, its ultimately impact, resource limitation and on the other hand, ultimate benefits of the rich biodiversity forced people to save their natural resources. Biodiversity action plan of the Pakistan should be exercised through government agencies and opportunities of alternative employability should be focused to overcome anthropogenic activities. Efficient and environment friendly urban settlement may promote through access of all possible necessities and allocation of industrial areas as well as protection of forests and agricultural lands ensure accordingly. 

4.Pl elaborate discussion

Future Conservation

Existence of plant biodiversity along the river Chenab is not only hosting present day food, fodder, fuel, timber and medicinal plants but also to the wild relatives of cultivated plants which may possess useful stress and disease resistant characteristics such as Desmostachya bipinnata L., Poa annua L., Saccharum arundinaceum Retz., and Saccharum spontaneum L. Simply these are wild germplasm or genetic resources. Moreover, there are species which are not important economically but have a significant ecological role in controlling flood and soil erosion as by Adhatoda vasica L. Conservation of biodiversity and their sustainable use in the scenario of climatic change can only be possible if the important species and habitats are carefully managed, examined and preserved like Bombax ceiba L. Summarizing the overall description for future management of the river Chenab two broad strategic approaches could be implemented to preserve ecological diversity. Firstly, stop illegal consumption of natural resources and secondly, restoration of plant biodiversity through reforestation, designating protected areas and multiplying rare species locally. Another point is provision of basic health care facilities to the local people to conserve over exploitation of medicinal plants. This study could be a baseline source of data for the management and conservation planning of the river Chenab and its catchment area. The solution to the problems is also addressed through following Biodiversity action plan of Pakistan too and struggle for achieving sustainable development goals. [73].

 Present studies recommends designation of new protected areas like riverine area of river Chenab and Bella Forest. Further, launching nurseries would also be a timely contribution to protect local flora and in broader spectrum, establishment of satellite seed and/or gene bank could also help in achieving conservation goals.

5. Discussion is very poor pl improve it and correlate it with your data

583. Site 2 parking area and land space was being used for construction of commercial buildings that affect plant biodiversity and at site 4 wild area had been converted to urban settlements, because of that plant species diversity had lower value then other sites.

591. Due to low anthropogenic activities wild core area in the surroundings of river is still intact. In soil, available potassium had maximum value than other sites that shows available potassium has significant role in alpha diversity.

593. Soil structure varies from site to site

594. All physical and chemical parameters of the soil are important for alpha diversity. 

596. Loamy soil has significant role in alpha diversity due to that it has maximum plant diversity than other soil structures.

605. These reporting anthropogenic activities affect the overall biodiversity which was depleting natural resources. 

609. Site 1 Jammu Tavi Bajwat area, located near the Bajwat wildlife sanctuary, this area is a game reserve where human intervention is too low. Biodiversity of the area was intact than other sites and its location referred it as the foothill of Himalaya region.

6.Ethics statement must be at end of the manuscript or do needful as per guideline of PLOS ONE

Needful done as suggested

7.Author are suggested to revisit and remove repeating of sentences .

Unnecessary text and ambiguity in the sentences have been removed, MS is proof read properly and text have been improved where deem fit. 

---

## [Editor Report · Decision Letter 1]

25 Jul 2022

Plant species diversity assessment and monitoring in catchment areas of river Chenab, Punjab, Pakistan

PONE-D-21-39895R1

Dear Dr. Iqbal

We’re pleased to inform you that your manuscript has been judged scientifically suitable for publication and will be formally accepted for publication once it meets all outstanding technical requirements.

Kind regards,

Tunira Bhadauria, Ph.D.

Academic Editor

PLOS ONE

Additional Editor Comments (optional):

The authors have done an excellent job of revising the manuscript. It gives me great pleasure to see the revised manuscript, which is now suitable for publication in the journal on a scientific level. The authors have taken into account all of the reviewers' suggestions and comments, and they have individually addressed each one. As a result, I recommend the journal to accept the work for publication.
---

## [Editor Report · Acceptance letter]

4 Aug 2022

PONE-D-21-39895R1 

Plant species diversity assessment and monitoring in catchment areas of river Chenab, Punjab, Pakistan 

Dear Dr. Iqbal:

I'm pleased to inform you that your manuscript has been deemed suitable for publication in PLOS ONE. Congratulations! Your manuscript is now with our production department. 

Kind regards, 

on behalf of

Dr. Tunira Bhadauria 

Academic Editor

PLOS ONE